# Exercise Restores Hypothalamic Health in Obesity by Reshaping the Inflammatory Network

**DOI:** 10.3390/antiox12020297

**Published:** 2023-01-28

**Authors:** Lucio Della Guardia, Roberto Codella

**Affiliations:** 1Department of Biomedical Sciences for Health, Università degli Studi di Milano, 20133 Milan, Italy; 2Department of Endocrinology, Nutrition and Metabolic Diseases, IRCCS MultiMedica, 20138 Milan, Italy

**Keywords:** brain–muscle–gut axis, metabolic disturbances, energy balance, physical activity benefits, adipose tissue

## Abstract

Obesity and overnutrition induce inflammation, leptin-, and insulin resistance in the hypothalamus. The mediobasal hypothalamus responds to exercise enabling critical adaptions at molecular and cellular level that positively impact local inflammation. This review discusses the positive effect of exercise on obesity-induced hypothalamic dysfunction, highlighting the mechanistic aspects related to the anti-inflammatory effects of exercise. In HFD-fed animals, both acute and chronic moderate-intensity exercise mitigate microgliosis and lower inflammation in the arcuate nucleus (ARC). Notably, this associates with restored leptin sensitivity and lower food intake. Exercise-induced cytokines IL-6 and IL-10 mediate part of these positive effect on the ARC in obese animals. The reduction of obesity-associated pro-inflammatory mediators (e.g., FFAs, TNFα, resistin, and AGEs), and the improvement in the gut–brain axis represent alternative paths through which regular exercise can mitigate hypothalamic inflammation. These findings suggest that the regular practice of exercise can restore a proper functionality in the hypothalamus in obesity. Further analysis investigating the crosstalk muscle–hypothalamus would help toward a deeper comprehension of the subject.

## 1. Introduction

Obesity increases the susceptibility to cardio-metabolic and neurodegenerative diseases, representing a major public health concern and an economic burden [1]. An array of molecules coordinates an intricate network regulating whole-body inflammatory/metabolic balance [2,3,4]. Overnutrition and the excessive accumulation of white adipose tissue (WAT) affect systemic inflammatory status and disrupt gluco-metabolic homeostasis by altering the functionality of central and peripheral tissues, including the hypothalamus [3,5].

The hypothalamus is key to integrating hormonal, environmental, and neural signals in the effort to provide adequate outputs to properly modulate feeding behavior and energy expenditure [6]. Obesity-associated pro-inflammatory molecules such as saturated fatty acids (sFFAs), tumor necrosis factor (TNFα), and advanced glycated end products (AGEs) activate microglia in the hypothalamus [7,8]. Chronic inflammatory response causes the disruption of internal hypothalamic circuitry, modifying hypothalamic outputs to other brain regions and the periphery, with negative consequences on satiety signal and energy homeostasis [8,9]. Conversely, the correction of such inflammatory/dysfunctional background warrants systemic metabolic homeostasis in obese animals and humans [10,11]. Alterations of neuronal activities in the mediobasal hypothalamus (MBH) compromise the capacity to correctly balance energy intake and expenditure, with long-term effect on body weight and fat mass [8,12].

Exercise represents an important challenge for organs regulating whole-body homeostasis [13], inducing several metabolic adaptions with positive impact on systemic health. Training optimizes energy substrates oxidation and promotes the secretion of array of muscle-derived biomolecules that counteract inflammation [5]. Thus, exercise emerges as a strategy for the management of a wide range of chronic diseases having inflammation as common soil [14,15,16,17]. Training-induced metabolic improvements are associated with cellular and molecular changes in central and peripheral tissues, all of which contribute to improve systemic insulin sensitivity and cardiometabolic health [18]. Recent evidence suggests that the positive effects exerted by exercise on systemic metabolic health can be in part mediated by the hypothalamus [13]. Thereby, this curbs obesity-associated hypothalamic inflammation in people affected by obesity is important to restore energy and metabolic homeostasis.

This review discusses animal and human evidence to highlight the positive effect of regular exercise on obesity-induced hypothalamic inflammation/dysfunction. This remodulation is hypothesized to activate a systemic anti-inflammatory network, favorable for the hypothalamus.

## 2. Inflammatory and Metabolic Consequences of Obesity and Physical Inactivity

WAT stores the energy surplus in the form of triglycerides (TG) [4]. Physical inactivity and nutrition excess typify the background conditions leading to TG over-storage, with consequent WAT expansion and development of insulin resistance [19,20]. When exposed to energy excess, white adipocytes undergo morphological and biochemical changes promoting cell dysfunction and the development of local inflammation [4,21]. TG accumulation and adipocyte volume expansion increase oxygen requirements and induce mechanical compression of blood vessel, producing hypoxia [21,22], which represents a critical stimulus for a switch toward a pro-inflammatory phenotype of WAT [2,4,21,23,24]. The activation of local and newly recruited inflammatory cell fuels the inflammatory process and the development of insulin resistance and mitochondrial dysfunction in WAT [4]. The release of adipocyte- and macrophage-derived mediators, such as TNFα and free fatty acids (FFAs) into circulation, increases the level of systemic low-grade inflammation, contributing to alter the functionality of central and peripheral tissues, critical for metabolic regulation [4]. In contrast to WAT, brown adipose tissue (BAT) converts TG into heat [21] via the sympathetic-activated uncoupled protein response (UCP)-1, in both animals and humans [21]. White and brown/beige adipocytes can mutually switch phenotype to meet systemic energy balance [25]. Higher energy demands stimulate adipocyte browning [21,25], while obesity and energy excess promote the conversion of brown adipocytes toward the white phenotype. A decreased mitochondria and impaired UCP-1 expression [21] characterize the white-shift in brown adipocytes, typically observed in animals and humans with obesity [21].

Skeletal muscle plays a pivotal role in the regulation of systemic inflammation and metabolic health [18], and variations in the oxidative efficiency of skeletal muscles contribute to the pathogenesis of obesity [18]. In obesity/inactivity, myocytes become dysfunctional and demonstrate signs of inflammation [18,24]. WAT-released pro-inflammatory mediators, such as FFAs and TNFα, play a critical role in inducing myocyte inflammation and dysfunction, in obesity [26,27]. The overexpression of the nuclear factor kappa-light-chain-enhancer of activated B cells (NF-κB), TNFα, and the accumulation of macrophages are typical features of obese subjects’ skeletal muscle [3,27,28,29,30]. Importantly, this setting correlates with the degree of insulin resistance and low-grade systemic inflammation [27,31]. Untrained and dysfunctional muscles, in obesity, contribute to alter whole-body inflammatory/metabolic balance by decreasing systemic glucose and lipid clearance and limiting the production of anti-inflammatory/immunoregulatory mediators contrasting systemic inflammation [3,24].

## 3. Obesity-Induced Hypothalamic Inflammation and Its Metabolic Consequences: Evidence from Animal Studies

The MBH regulates BAT and WAT metabolism and governs energy balance [32,33,34]. Within the MBH, specific neuronal subpopulations in the arcuate nucleus (ARC)—pro-opiomelanocortin and agouti-related protein neurons (POMC and AgRP, respectively)—control, with opposite actions, food intake and energy expenditure [35]. Leptin exerts part of its anorexigenic effect by activating the Janus-activated kinase (JAK) signal transducer and the activator of transcription 3 signaling (STAT3) in POMC neurons. This produces the synthesis of α-Melanocyte-stimulating hormone (MSH), which mediates anorexigenic and thermogenetic effects. On the other hand, the activation of AgRP neurons increases the orexigenic stimuli and lowers energy rates, partly by activating other hypothalamic areas. Alterations of neuronal activities in the ARC compromise the capacity to properly balance energy intake and expenditure, resulting in negative long-term effects on body weight and fat mass [12].

Obesity and overnutrition induce chronic low-grade inflammation in the MBH, primarily in the ARC nucleus [7]. High fat diet (HFD)-fed mice display an upregulation of the main pro-inflammatory pathways—c-Jun N-terminal kinase (JNK) and NF-κB—along with an overproduction of pro-inflammatory mediators (e.g., TNFα) [7] and microglial activation [36,37]. Microglia are macrophage-like cells of the central nervous system, which are activated by pro-inflammatory signals, causing local production of pro-inflammatory cytokines. Importantly, gliosis has been identified also in obese humans [36,38], and it strongly correlates with BMI and peripheral insulin resistance, suggesting the presence of similar adaptive mechanisms demonstrated in mice.

WAT-released pro-inflammatory molecules can mediate the inflammatory effect on the hypothalamus [8,36,39]. In obesity, the suppression of insulin signal and inflammation in WAT enhance the chronic release of FFAs and other pro-inflammatory mediators [40]. WAT-released FFAs, for example, can play a key role in the induction of inflammation and insulin resistance via the activation of the toll-like receptor (TRL)-4/NF-κB/JNK signalling in the hypothalamus [8]. In addition, WAT-derived macrophages have been demonstrated to infiltrate the MBH—in obese animals [41]. Notably, hypothalamic inflammation in animal models precedes the development of obesity and WAT dysfunction, suggesting that overnutrition itself has a critical role in this process [8]. Saturated FFAs (sFFAs) promote microgliosis by activating the TLR4-Myd88 and ceramide-dependent pathways [7,36]. Likewise, increased levels of advanced glycated end-products (AGEs), following a high carbohydrate diet, were shown to activate microglial response in the hypothalamus in mice [42].

The development of chronic inflammation in the hypothalamus has detrimental effects on energy expenditure and body weight (Figure 1) [6]. Hypothalamic inflammation induces leptin and insulin resistance [8,43,44], disrupting hunger response, lowering energy expenditure [8,45], and inducing weight gain in animals [8,45,46]. Conversely, the inhibition of the pro-inflammatory pathway inhibitor of nuclear factor kappa-B kinase (IKKβ)/TLR-4 downstream factor MyD88 prevents HFD-induced hypothalamic insulin and leptin resistance, effectively contrasting excessive food intake and weight gain in mice [47,48,49]. Dysfunction in the ARC induces the decline in voluntary locomotor activity [6] and associates with lower energy expenditure rates in mice [46]. The stimulation of leptin-STAT3 pathway in HFD animals in the ARC and ventro-medial hypothalamic (VMH) nuclei was significantly reduced compared to lean littermates [50]. Notably, the suppression of microglia-induced inflammation in the ARC was sufficient to: (a) recover leptin sensitivity; and (b) limit food intake and WAT accretion in HFD-fed rodents [51].

## 4. Exercise and Obesity-Induced Hypothalamic Inflammation

Exercise-induced adaptive response in the hypothalamus leads to the improvement of insulin and leptin signaling in the ARC and positively modulates the activity of POMC and AgRP neurons, resulting in decreased food intake and body weight control [13].

### 4.1. Animal Studies

Findings from animal models suggest that exercise ameliorates obesity-associated hypothalamic inflammation and dysfunction (Table 1). In obese HFD-fed mice, an acute session of swimming (two exercise bouts of 3-h each, interleaved by 45-min resting periods) improved endoplasmic reticulum (ER) stress markers (e.g., TRB3), as well as adiponectin and insulin signalling [52] in the hypothalamus. Similarly, an acute session of running (three bouts of 45 min with 15 min of recovery at 60% of the exhaustion velocity) was capable to restore the central anorectic effect insulin in the hypothalamus of obese mice [53]. In the study of Yi et al., a six-month moderate-intensity treadmill running (30 min, 5 m/min for five times/week, 10% inclination) was sufficient to prevent HFD-induced hypothalamic inflammation by decreasing obesity-induced microglia activation in the ARC [54]; importantly, the observed decline in microglial activation was associated with a significant improvement in glucose tolerance and insulinemia. In a group of HFD-fed obese animals, both acute treadmill running (60 min, 10–15 m/min at a 5% inclination) or repetitive sessions of swimming (60 min/day, five times/week, for four weeks with an overload of 2.0% of the body weight) were capable to reduce IKKβ and IKBα phosphorylation and markers of ER stress in the hypothalamus [55], along with a restoration of the NPY and POMC mRNA levels. This effect was associated with improved insulin and leptin sensitivity in the hypothalamus, resulting in decreased food intake [55]. Similarly, Wang et al. demonstrated that microgliosis and markers of inflammation (e.g., IL-1β, TNFα, and NF-κβ) were significantly ameliorated in HFD-mice undergoing eight weeks of swimming (50 min, daily) [56]. In two different groups of trained and control animals, twelve weeks of HFD significantly increased cell apoptosis in the ARC; notably, the group undergoing voluntary running resulted partly protected from the HFD-induced apoptotic effects [50]. In obese mice, 20 days of swimming (five days/week) effectively decreased the concentration of transforming growth factor-beta 1 (TGF-β1) and nuclear factor of kappa light polypeptide gene enhancer in B-cells inhibitor-α (Ikβ-α) in the hypothalamus [57]. This effect paralleled with an increase in energy expenditure and a reduction in food intake and body weight. Likewise, a three-month voluntary training was capable to partially restore leptin-induced STAT3 phosphorylation in HFD-fed mice, suggesting that exercise improves central leptin signaling in obesity [50]. Additionally, mid-term voluntary running was capable to restore the number of POMC neurons, reducing the HFD-induced apoptosis in POMC-expressing neurons in the hypothalamus [50]. Exercise was shown to activate leptin receptor-positive neurons in the VMH. In the study of Krawczewski Carhuatanta et al., protein fosB (FosB) immunoreactivity in the VMH was decreased in HFD-fed inactive mice, while a repairing effect was reported in voluntary-running mice, suggesting the actual recovery of neuronal function [58]. Interestingly, this effect was independent from the body fat percentage, which evidences the existence of an alternative mechanism through which regular exercise can reestablish hypothalamic functionality [58]. To further suggest the critical activity of exercise on hypothalamic inflammation, the intraventricular administration of leptin was shown to be ineffective in decreasing food intake and body weight in inactive mice exposed to HFD [58]. Conversely, central leptin injection was followed by a significant decrease in body weight only in HFD-fed mice on voluntary training [58], strongly suggesting a leptin-sensitizing effect of chronic exercise. Eight weeks of wheel running (60 min, five day/week, at 60% of maximal workload) increased leptin signaling (JAK2/pSTAT3) and reduced the content of markers of inflammation (e.g., TLR-4, IkKβ and others) kinase in the hypothalamus of obese mice [59]. Such results were paired by a reduction in ER stress and a macrophage activation. Furthermore, wheel running decreased the expression of the proapoptotic protein (PARP1) and increased the expression of IL-10 and the anti-apoptotic proteins Bcl2, preserving pro-opiomelanocortin mRNA expression [59]. Similarly, an intense running (initial speed of 15 m/min gradually increased to 35 m/min, 20–25 min, for seven weeks) increased POMC mRNA levels in the hypothalamus [60] and regular exercise activated the POMC, reducing food intake and body weight. The activation of POMC is likely secondary to an exercise-induced remodeling effect in synaptic connections in leptin-receptor expressing neurons in this area [61].

### 4.2. Human Studies

In accordance with animal evidence, early findings in humans show changes of circulating levels of NPY/α-MSH in trained overweight and obese individuals [62,63], suggesting an exercise-induced hypothalamic improvement. In a randomized-controlled trial carried out in a group of older adults, both low resistance and high resistance training (40 and 80% of one repetition maximum, respectively), performed over 12 weeks, resulted in a significant drop in circulating NPY levels [63]. Other studies including multi-intervention protocols (exercise, diet, and others) have evidenced some beneficial effects on hypothalamic health in obesity. In obese adolescents, for example, a mixed training type (aerobic + resistance exercise; aerobic training: 30 min, three times/wk at first ventilatory threshold; resistance training: 30 min, three times/wk, three sets of 15–20 repetitions; volume and intensity were adjusted inversely to decrease the number of repetitions from 15–20 to 10–12 and six to eight), along with lifestyle changes, led to a decrease in body adiposity and AgRP levels after six months compared to baseline. The study also demonstrated that α-MSH levels increased in the aerobic exercise-trained group after 12 months [62]. Similarly, in obese and metabolic syndrome-affected adolescents, one year of combined diet + exercise (aerobic + resistance exercise) procured a significant decrease in fat mass, NPY levels (0.94 CI [0.43–2.25] vs. 1.19 CI [0.55–2.3]) and AgRP/NPY ratio (0.71 CI [0.36–1.77] 0.57 CI [0.27–1.86]) [64]. A multi-intervention protocol (exercise + diet + psychotherapy) showed a drop in NPY/AgRP ratio and its negative correlation with fat mass in a group of adolescents following a long-term mixed exercise, diet, and behavioral therapy [65]. Altogether, this evidence underpins a positive regulatory effect of long-term training on hypotalamic functionality in humans. However, although being indicative, the peripheral assessment of NPY/AgRP/MSH represents a surrogate via to assess hypothalamic functional improvement. More reliable and in-depth measures on hypotalamic inflammation/functionality are required to support current evidence.

## 5. Exercise and Obesity-Induced Hypothalamic Inflammation: Pathophysiological Mechanisms

Different mechanisms can be addressed to explain the positive effects of exercise on the hypothalamus in obesity, although the exact pathophysiology in still poorly explored. Exercise-produced cytokines and mediator are able to cross the brain–blood barrier (BBB), regulating vascularization, neuronal plasticity, and inflammation in diverse areas of the brain [13]. Changes in regulatory cytokines across the (BBB) can be critical in regulating neuroinflammation and brain health in dysmetabolic condition [8]. The release of anti-inflammatory and immune-modulatory cytokines from skeletal-muscle reduces ER stress and improves insulin and leptin sensitivity [13].

### 5.1. Exercise-Induced Mediators Reduce Hypothalamic Inflammation in Animal Models

Skeletal muscle is a secretory organ that responds to either exercise or inactivity by releasing myokines mediating the communication of the skeletal muscle with central and peripheral organs [18].

Interleukin (IL)-6 is a cytokine presenting a bimodal activity, being implicated in both pro-inflammatory and anti-inflammatory functions, depending on the differential signaling pathways activated in targeted tissues [18,66]. IL-6 has been shown to be involved in mediating hunger suppression, WAT lipolysis, and body weight reduction, regulating systemic metabolic health [18]. Muscle-derived IL-6 stimulates lipolysis [67,68,69] and FFA oxidation in WAT [69,70]

Recent studies suggest a role of muscle derived IL-6 in the mitigation of inflammation and metabolic dysfunction in the hypothalamus (Figure 2) [55], which would partly account for the positive effects of acute IL-6 secretion on whole-body metabolism [18]. IL-6 can cross the BBB, regulating hypothalamic nuclei and circuitry controlling hunger/satiety and energy expenditure. IL-6 receptor is expressed in microglia, ependymocytes, endothelial cells, and astrocytes [71]. The administration of exogenous IL-6 induces neurogenesis-related gene expression in neuroprogenotor cells (NPCs) and in the hypothalamus of mice [71]. IL-6 improves glucose tolerance and suppresses food intake when centrally administered, and mice lacking muscle IL-6 were found to scarcely respond to leptin administration [71].

IL-6 concentrations augment after a single session of exercise [18]. In exercised muscles, IL-6 secretion is stimulated by augments in cytosolic Ca^2+^ and calcineurin via a p38 mitogen-activated protein kinase (p38MAPK)-mediated mechanism [72]. Interestingly, an acute increase in IL-6 post-exercise has been demonstrated also in the hypothalamic milieu and in microglia, astrocytes, and neurons of different areas of the brain of exercised mice, suggesting either BBB cross or local increased expression, following exercise [55,73]. Increased hypothalamic IL-6 expression would contrast with the HFD-induced activation of the IKKβ/NF-κB pathway, thus preventing HFD-induced neuronal inflammation/apoptosis and neuronal loss [55]. In favor to this assumption, it was shown that the inhibition of the hypothalamic-specific IL-6 suppressed the beneficial effects of exercise on the re-balance of food intake and insulin and leptin resistance in mice [55]. A recent study demonstrates that exercise-induced IL-6 activates the phosphorylation of JAK2/Tubby protein (TUB) in the hypothalamus, significantly reducing food intake [74]. Other evidence suggests that IL-6, after acute exercise, can control the sphingosine-1-phosphate receptor 1 (S1PR1)–mediated activation of STAT3 in the hypothalamus [75]. Indeed, both acute exercise and the intra-ventricular IL-6 injection increased S1PR1 levels and STAT3 phosphorylation in the hypothalamus of both lean and obese mice. Importantly, this event was accompanied by a significant decrease in food intake, supporting that the metabolic-regulatory effects of exercise are partly mediated by the restoration of insulin and leptin signalling, as well as involving IL-6 signalling [75]. In line with this hypothesis, exercise failed to stimulate the S1PR1-STAT3 signaling in IL-6-ablated mice. Similarly, the disruption of hypothalamic-specific IL-6 action stemmed the inhibitory effects of exercise on food intake and lowered S1PR1 protein content in the hypothalamus, indicating that the activity of IL-6 in the hypothalamus is critical for eliciting the positive systemic metabolic effects of exercise [75].

IL-10 possesses systemic anti-inflammatory effects. IL-10 inhibits the production of TNFα by immune cells [76], elicits a suppressive activity on MHC-II [77], and mitigates its activation through inhibiting cathepsin S via activating the STAT3 signalling [78]. IL-10 expression is increased in trained muscles of exercised rodents [79], and its plasma levels typically increase in healthy and diabetic humans after prolonged exercise [5,80,81].

IL-10 can cross the BBB; its levels have been found significantly augmented in the hypothalamus of HFD-fed mice undergoing chronic exercise [59]. IL-10 has been shown to be critical in the control of obesity-induced neuroinflammation, warranting the proper functionality of POMC/AgRP neurons in rodents. The overexpression of IL-10, obtained via a viral vector, was sufficient to suppress the HFD-induced activation of IKKs and SOCS3 and restored POMC expression in ARC of obese mice [82]. Interestingly, this effect was paralleled by a diminished WAT accumulation, suggesting a crucial role of hypothalamic IL-10 in the regulation of energy balance and body weight. Interestingly, the hypotalamic regulatory effects of IL-6 seem to be partly mediated by the anti-inflammatory cytokine IL-10 [55]. To corroborate this hypothesis, Ropelle et al. demonstrated that exercise failed to reverse the pharmacological activation of IKKβ and ER stress in the hypothalamus in mice with suppressed IL-6 and IL-10 signaling [55]. Therefore, the cooperation between IL-6 and IL-10, in the hypothalamus can generate an effective anti-inflammatory mechanism contributing to: (i) the recovery of neuronal function and insulin and leptin signalling; (ii) re-establishing the anorexigenic and energy-regulatory effects disrupted by obesity and overnutrition. This cooperative activity is further supported by the evidence that exercise-induced muscle IL-6 exerts a systemic anti-inflammatory response through bosting systemic IL-10 increase [18,83,84].

Other exercise-induced mediators seem to have favorable effects in regulating inflammation in the hypothalamus, with effects on energy balance. Early observations suggest that β-aminoisobutyric acid (β-AIBA), for example, can play a role in this respect. β-AIBA is an exercise induced muscle-released mediator [85], which is believed to mediate part of the regulatory effects of exercise on cardiometabolic status by attenuating obesity-associated inflammatory response in central and peripheral organs and increasing the browning of adipose tissue [86,87]. Interestingly, Park et al. demonstrated that β-AIBA administration was capable of reverting palmitic acid-induced hypothalamic inflammation and microglial activation in mice, contrasting WAT accumulation and weight gain in HFD mice [88].

### 5.2. Pro-Inflammatory Mediators Disrupt Hypothalamic Health

Obesity-associated pro-inflammatory molecules (e.g., TNFα, FFAs, Resistin, AGEs, and CCL-2) can cross the BBB, propelling hypothalamic inflammation (Figure 2) [7,8,89,90].

TNF-α elicits negative effects on neurogenesis and mitochondrial function in the brain [91]. TNFα binding to its receptors (TNFR1 and 2) activates both NFκB and STAT3 signaling pathways in the hypothalamus [92]. This leads to mitochondrial and ER dysfunction and increased oxidative stress [91,92], promoting hypothalamic insulin and leptin resistance [7,93]. In NPY/AgRP neurons, TNF-α significantly upregulated IκBα, nuclear factor (NF)-κB [7]. TNFα treatment impaired mitochondrial function, increased ROS production, and decreased the expression of pro-neurogenic protein (Mash1/Ngn3) in hypothalamic NPCs [91]; these results paralleled with and increased AgRP protein expression and a decline in POMC.

AGEs promote systemic inflammation [94], and their levels are typically higher in obesity and dysmetabolic conditions [95]. AGEs bind to their specific receptors (RAGEs) or to non-specific receptors, such as CD36 and AGE receptor-1 [96], located on a variety of cell types including immune cells and microglia. The binding of AGEs activates the nuclear Ras–mitogen-activated protein kinase (MAPK) and NF-κβ pathways [96]. In the hypothalamus, AGEs have been shown to induce microglial response and have a pivotal role in the build-up of inflammation in HFD-fed animals [42]. The increase in AGEs following a high carbohydrate diet-induced was found to induce hypothalamic inflammation in rodents [42], whereas a very low-carbohydrate diet mitigated microglia proliferation in HFD-fed mice [51]. The high-fat and high-carbohydrate diet also caused an increase in N-carboxymethyl lysine immunoreactivity in both POMC and NPY neurons [42]. Animals lacking RAGEs exhibit an improved metabolic phenotype and decreased microglial reactivity on a HFD [42], which confirms the prominent role of AGEs in inducing hypotalamic dysfunction.

Resistin can play a role in regulating hypothalamic inflammation. While in mice it is mainly secreted by WAT, in humans a significant portion of resistin is produced by immune cells [43]. Resistin is typically elevated in animals and humans with obesity [97] and is thought to activate the inflammatory pathway TLR-4-JNK, inducing insulin resistance in the hypothalamus [98]. In addition, resistin controls the expression of leptin receptor and suppressor of cytokine signaling 3 (SOCS-3) in the ARC [99], suggesting an inflammatory activity in this brain region.

Dietary and WAT-derived FFAs can activate the microglia and astrocytes through stimulating the TRL-4 pathway and stimulating the production of ceramides [100], ultimately producing inflammation in the hypothalamus [101]. Recent studies have identified lipid-sensing G-protein-coupled receptors in the hypothalamus, revealing their involvement in the regulation of energy balance, as well as in the hypothalamic inflammatory response that occurs in obesity [101]. Long chain FFAs were found to inhibit POMC neurons and activate NPY/AgRP neurons [102]. Due to the critical role of FFAs in inducing inflammation in different MBH nuclei [8], a diminished flow of FFAs toward the hypothalamus can contribute to mitigate microglial response, inflammation, and insulin resistance in this brain area [8,101].

CCL-2 is a chemotactic factor, which plays a critical role in monocyte infiltration in central and peripheral tissues in obesity [103]. In obesity, CCL-2 is mainly produced by WAT, and its levels correlate with the degree of WAT accumulation in animals and humans [4,103,104]. In murine models, CCL-2 has been shown to facilitate the LPS-induced inflammatory response in the hypothalamus [89] and is involved in the infiltration of WAT-derived macrophages in the brain [41].

#### 5.2.1. Effect of Exercise on Pro-Inflammatory Mediators

##### Animal Studies

Both acute and chronic exercise interventions have been shown to control TNFα levels. TNFα expression significantly decreased in myocytes of rats trained on a treadmill (13–20 m/min for 60 min/day, five days/week, for 10 weeks) [79]. Swimming (five times per week for six weeks) significantly reduced TNFα levels in HFD-fed rodents [105]. Regular exercise lowered TNFα release from immune cells [106] and suppressed TNFα in the hypothalamus of rats affected by cancer-induced cachexia [107]. As above evidenced, the improvement of WAT function, following exercise, accounts for TNFα levels drop [108,109,110,111]. In addition, muscle-derived myokines seem to play a role in modulating TNFα levels [18]. Exercise promotes the release of IL-1 receptor antagonist (IL-1ra) [112,113], which was demonstrated to contrast the secretion of IL-1α and TNFα, in vitro [114].

Regular exercise ameliorates glycemic control and lowers low-grade inflammation [115], helping toward the control of circulating AGEs and preventing their formation [116]. The increased oxidative activity during and after exercise reduces the availability of glycating intermediates deriving from glycolytic (e.g., glucose-6-phosphate, fructose-6-phosphate) and polyol (e.g., fructose-3-phosphate, 3-deoxyglucosone) oxidative pathways [15,117,118]. In HFD-fed rodents, for example, moderate-intensity chronic wheel running (five-week accommodation phase with increasing exercise intensity 15 m/min for 30, 45, and 60 min, respectively; fourth and fifth week: 20 m/min for 30 and 45 min, followed by a five-week constant training period at 20 m/min for 60 min), was shown to efficiently decrease circulating AGEs [117,119].

Early evidence demonstrates that exercise can lower resistin levels in animal models of obesity. In obese/dysmetabolic mice, for example, exercise was observed to downregulate resistin via suppressing MALAT1 and activating miRNA-382-3p expression [120].

Both regular training and acute exercise were shown to have positive effects on CCL-2 levels [105,121,122,123]. In HFD-fed rats, for instance, swimming (60 min a day, five times per week for six weeks) produced a significant decrease in CCL-2 [105].

To note, since dysfunctional WAT represents the principal producer of circulating pro-inflammatory mediators [8,41,101], the regulation of energy balance and TG storage in white adipocytes underlies part of the anti-inflammatory effect of exercise [123]. In rodents’ WAT, exercise reduces macrophage accumulation [19] and promotes macrophages shifting toward the anti-inflammatory (M2) phenotype [124]. Muscle-derived mediators such as IL6, Brain-derived neurotrophic factor (BDNF), and meteorin-like protein (METRNL) can restore a proper functionality in WAT by augmenting lipid oxidation, mitigating inflammation [4,125,126]. The reduction of WAT mass, following training, is associated with lower levels of circulating pro-inflammatory mediators in obese animal and humans [108,109,110,111].

##### Human Studies

In humans, a 3-h cycling exercise at 75% of maximal aerobic capacity (VO_2max_) significantly lowered TNFα levels upon lipopolysaccharide (LPS) stimulation [127], and 12 weeks of either light or vigorous resistance exercise (respectively 40% and 80% of one repetition maximum) exercise led to a significant reduction in TNFα levels in older subjects affected by obesity [63]. In humans, exercise-driven IL-6 and IL-10 suppress the secretion of TNFα, inducing a drop in its plasma levels [76,127,128].

Regular exercise is capable to lower plasmatic FFAs, in obesity [109,129], augmenting their clearance from bloodstream [109]. In overweight and obese individuals, for example, a significant drop in FFAs levels after six weeks was demonstrated (20 min; three times/week; exercise at 60 to 85% of VO_2max_), [130] and three months of aerobic training (60 min/day at 65% of maximal oxygen uptake) plus diet [109] (Table 2), as result of increased muscle FFA oxidation [109]. As for TNFα, the exercise-induced reduction of WAT mass is associated with lower concentrations of FFAs in obese-dysmetabolic subjects [108,109,110,111], strongly suggesting that the amelioration of WAT status partly mediates the effects in FFAs levels.

The practice of physical activity is correlated to increased soluble AGEs receptor (sRAGE) and a lower AGE/sRAGE ratio in type-2 diabetic individuals [131]. Consistently, in a randomized clinical trial on obese-dysglycemic individuals, 12 weeks of supervised aerobic exercise (five days/week, 60 min/day at 65–85% Heart Rate_max_) along with dietary counselling, efficaciously lowered sRAGE plasma concentration, suggesting a protective activity of physical activity, in obesity [118].

Resistin was demonstrated to decrease in humans undergoing both resistance and endurance training [132]. In overweight/obese individuals, both 16 weeks of aerobic exercise (45–60 min sessions per week at 50–85% maximum oxygen consumption [VO_2max_]) [81] and 12 months of moderate exercise (two times/week) [133] significantly reduced resistin, independently from WAT mass variations [133].

In randomized-controlled trials in obese-dysmetabolic subjects, the regular practice of endurance exercise was shown to significantly lower CCL-2 plasma levels [111,123]. In morbidly obese adolescent women (BMI > 40), eight weeks of moderate-intensity exercise intervention (180 min/week at 40–55% VO_2max_) produced a significant decline in CCL-2 and resistin levels [111]. A 10% decline in CCL-2 from baseline levels was registered by Christiansen et al. in a group of obese individuals undergoing 12 weeks of moderate exercise [123]. Interestingly, these findings are consistent with evidence demonstrating that the exercise-induced reshaping of WAT functionality warrants a decrease in CCL-2 [19,121,122], suggesting that the modulation of WAT may be critical for regulating plasma CCL-2 levels in obesity.

**Table 2 antioxidants-12-00297-t002:** Effect of exercise-modulated mediators on the hypothalamus of animals and humans.

Study	Model	Type of Exercise/ Frequency/Intensity	Effect of Exercise	Effect on the Hypothalamus
Ropelle et al. [55]	HFD-mice	Running; one session; 60 min, 10–15 m/min at a 5% inclination	↑ IL-6	↓ IKKβ, IKβα [55]
Ropelle et al. [55]	HFD-mice	Running; one session; 60 min, 10–15 m/min at a 5% inclination	↑ IL-10	Suppression of IKKβ [55]
Wasinski et al. [105]	HFD-mice	Swimming; 60 min a day, five times per week for six weeks	↓ TNFα	↓ IKKβ, IKβα * [7]
Boor et al. [117]	Zucker rats	five weeks: 15 m/min for 30, 45, and 60 min; five-week constant training period at 20 m/min for 60 min	↓ AGEs	↓ Microglial activation * [42]
Wasinski et al. [105] Many et al. [111]	HFD-mice [105] Obese subjects	Swimming; 60 min a day, five times per week for six weeks Aerobic training; eight weeks, 180 min/wk at 40–55% VO_2max_	↓ CCL-2	↓ Inflammation * [89]
Solomon et al. [109] Shojaee-Moradie [130]	Overweight and obese subjects	Aerobic training; three months (60 min/day at 65% of VO_2max_ Aerobic training; six weeks; 20 min; three times/week; exercise at 60 to 85% of VO_2max_),	↓ FFAs	↓ Microgliosis inflammation * [134]
Kadoglu et al. [81] Gondim et al. [133]	Overweight and obese subjects	Aerobic training; 16 weeks of, 45–60 min sessions per week at 50–85% VO_2max_ Swimming or water aerobics; 12 months; 60 min, 2 times/week at 65% HR_max_	↓ Resistin	↓ Inflammation * [98]

↑ increase. ↓ reduction. AGEs, advanced glycation end- products; CCL-2 chemokine (C-C motif) ligand 2; FFAs, free fatty acids; IL, interleukin; TNFα, tumor necrosis factor-α. * The exercise-induced effect of these mediators is not directly demonstrated on the hypothalamus.

### 5.3. Effect of Exercise on the Gut-Hypothalamus Crosstalk

Overnutrition and obesity have detrimental effects on the gut. Obesity is associated with a pro-inflammatory switch of microbiota [135], lower short-chain fatty acids (SCFAs) production and increase mucosal inflammation and permeability in the gut [18]. Interestingly, in obesity, gut permeability increases independently of dietary changes [136], suggesting that systemic inflammation worsens gut health reshaping intestinal epithelial cells toward a dysfunctional phenotype [137]. The increased gut permeability allows the translocation into circulation of pro-inflammatory molecules such as LPS [137,138,139], which can pass the BBB and eventually activate microglia through the TLR-4-dependent pathway [140]. The prolonged exposure to LPS increases the phosphorylation of JNK in the hypothalamus of rodents and impairs insulin-induced AKT phosphorylation and the translocation of Foxo1 from the nucleus in hypothalamic cells, producing insulin resistance and inhibiting the satiety feedback [140].

While there is no current experimental evidence showing that the positive effect of exercise is mediated by the gut functional improvement, available findings suggest that exercise-induced gut-microbiota/gut permeability improvement can positively influence hypothalamic inflammation/functionality (Figure 3). The regular practice of moderate exercise was demonstrated to restore gut inflammation, improving its functionality in obese animals [18,141,142,143]. A large body of evidence suggests that moderate exercise can remodulate the gut microbiota towards an anti-inflammatory phenotype [141,144,145] promoting the abundance of species such as *Faecalibacterium prausnitzii* and *Akkermansia muciniphila* [145], whose presence is critical for the maintenance of a health intestinal ecosystem and lower systemic inflammation [141]. Interestingly, changes in microbiota after exercise seem to occur regardless of type of diet consumed [135], suggesting that exercise benefits are independent to nutritional changes. In HFD-fed mice, the constant practice of moderate exercise increases microbiota diversity and stimulates the growth of SCFAs-producing bacteria [143,144,145]. Gut-released SCFAs directly act on the hypothalamus: they cross the BBB, using a specific monocarboxylate transporter, whose activity is increased following acute exercise [146]. SCFAs provide energy for brain microglia and exert neuroprotective effects [147]. One of the most commonly secreted SFCAs—butyrate—was shown to stimulate neural proliferation in different brain regions, inducing neurogenesis in mice [148,149]. Importantly, according to recent evidence, butyrate is capable of reverting obesity-induced hypothalamic inflammation and microglial activation in mice models [150].

The restoration of a proper gut permeability contributes to lower the translocation of bacterial antigens and LPS into circulation [151], with positive effects on systemic and hypothalamic inflammatory state [143,152,153]. The regular practice of moderate exercise has proven to ameliorate gut-barrier integrity in both obese animals and humans, limiting LPS translocation into bloodstream [135,143]. The microbiota switch and the production of SCFAs are critical in this respect [147]. In HFD-fed mice, for example, the practice of voluntary wheel-running for four weeks successfully decreased plasma LPS levels [143]. Similarly, observational studies in humans demonstrated that trained subjects show lower concentration of plasma LPS compared to sedentary individuals [154]. In intervention studies on dysmetabolic individuals, exercise efficiently mitigated the translocation of LPS to bloodstream [152,153]. In the study of Motiani et al. [152], for example, both interval training (four-to-six bouts of 30-s each, of all out cycling efforts with 4 min of recovery) and moderate-intensity bicycling (40 to 60 min at 60% of VO_2max_) lowered endotoxemia in overweight-to-obese diabetic subjects.

Future studies examining in-depth the three-way relationship “exercise-gut-hypothalamus” are warranted to support this pathophysiological model.

## 6. Conclusions

Obesity and overnutrition induce inflammation and dysfunction in key hypothalamic nuclei governing energy homeostasis and metabolic health. The findings discussed in the present review suggest that moderate exercise is capable of improving hypothalamic inflammation in obesity, restoring leptin and insulin sensitivity, and exerting positive effects on food intake and body weight (Figure 3). Importantly, the reshaping activity on the hypothalamus is likely to mediate part of the beneficial effects of exercise on systemic energy balance in obesity. Further studies are needed to explore, in-detail, the muscle-brain crosstalk. Evidence linking inflammation and functional impairment in the hypothalamus would ensure a deeper pathophysiological interpretation of the effects elicited by exercise. Likewise, further notions elucidating how exercise-modulated mediators operate on the hypothalamus would be critical for a full understanding of the subject.

## Figures and Tables

**Figure 1 antioxidants-12-00297-f001:**
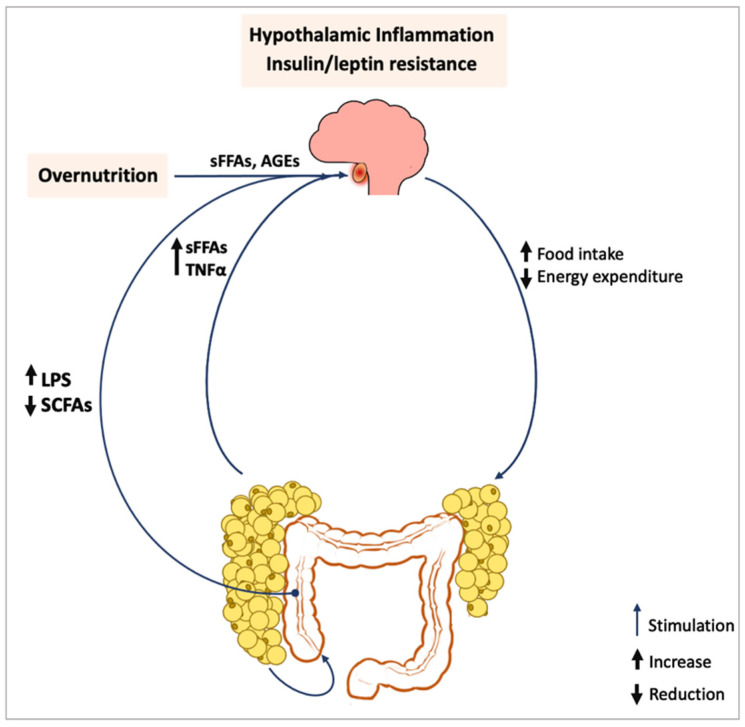
Obesity-associated disruption of hypothalamic functionality. AGEs, advanced glycation end-products; CCL-2 chemokine (C-C motif) ligand 2; LPS, lipopolysaccharide; SFCAs, short chain fatty acids; sFFAs, saturated fatty free acids; TNFα, tumor necrosis factor-α.

**Figure 2 antioxidants-12-00297-f002:**
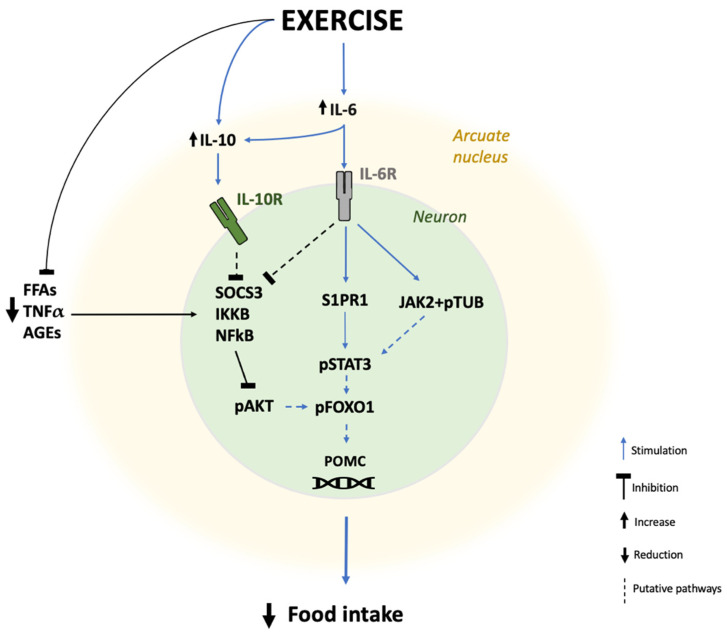
Potential pathways underlying the effect of exercise in restoring insulin and leptin and insulin sensitivity in the hypothalamus of obese animals and humans. Exercise-induced suppression of inflammatory mediators (e.g., FFAs, AGEs, TNFα), along with IL-6 and IL-10 increase, downregulate the inflammatory signalling, with positive effects on the insulin pathway. Exercise-induced IL-6 restores the leptin signalling (JAK2, pSTAT3), via activating S1PR1 and TUB, with the final effect of stimulating the transcription of POMC. AGEs, advanced glycation endproducts; AKT, protein kinase B; Foxo1, Forkhead box protein O1; IL, interleukin; JAK2, Janus kinase 2; IKKB, inhibitor of nuclear factor kappa-B kinase subunit beta; NF-κB, nuclear factor kappa-light-chain-enhancer of activated B cells; POMC, pro-opiomelanocortin; S1PR1, sphingosine-1-phosphate receptor 1; SOCS3, suppressor of cytokine signaling 3; STAT3, Signal transducer and activator of transcription 3; TNFα, tumor necrosis factor-α.

**Figure 3 antioxidants-12-00297-f003:**
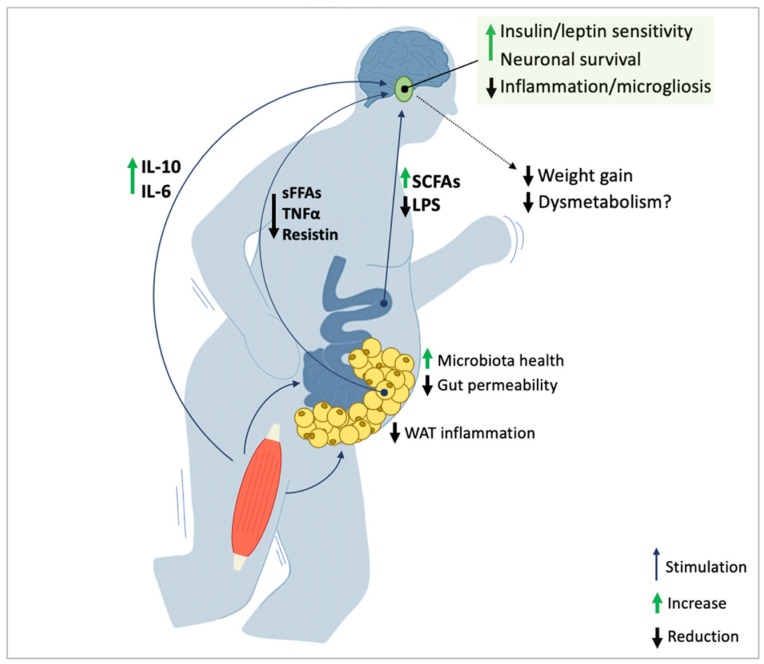
Exercise can modulate mediators involved in the regulation of hypothalamic health. Regular training suppresses microglial response and downregulates the main markers and pathways of inflammation (e.g., TNFα, NF-kβ and IKKβ) in the ARC and other energy-regulatory nuclei in the MBH. Exercise-induced anti-inflammatory response in the hypothalamus pairs with increased leptin and insulin sensitivity, improved neuronal survival in the ARC, and associates with lowered food intake and body weight control, which remarks the central role of remodulating inflammation to restore a proper hypothalamic functionality. The anti-inflammatory/regulatory effect of exercise on the hypothalamus is partly achieved through the increase in myokines such as IL-6 and IL-10. These cytokines mediate the inhibitory effect on hypothalamic inflammation enhancing leptin sensitivity and adjusting energy expenditure rates during/following exercise. The positive activity of regular exercise on WAT dysfunction along with the immunomodulatory effect of training are critical for the suppression of key mediators (e.g., FFAs, Resistin, AGEs) that produce inflammation and insulin/leptin resistance in the ARC. The exercise-induced switch in microbiota is responsible for augmented production of SCFAs, with potential neuroprotective and anti-inflammatory effects on the hypothalamus; the improvement of the gut barrier following regular exercise lowers the translocation of LPS, preserving the hypothalamus and other brain areas from the development of inflammation. CCL-2 chemokine (C-C motif) ligand 2; IL, interleukin; LPS, lipopolysaccharide; SFCAs, short chain fatty acids; TNFα, tumor necrosis factor-α; WAT, white adipose tissue.

**Table 1 antioxidants-12-00297-t001:** Effect of exercise on hypothalamic inflammation and dysfunction in rodent models of obesity.

Study	Model	Type of Exercise/Frequency/ Intensity	Effect on Hypothalamus	Metabolic Changes
De Almeira Rodriguez et al. [53]	HFD mice	Swimming; one session two bouts of 3-h each	↓ ER stress (TRB3) ↑ Foxo-1	↑ Central anorectic effect of insulin
Calais Gaspar et al. [52]	Leptin-stimulated HFD mice	Running; one session; three bouts of 45 min each at 60% of the exhaustion velocity	↓ ER stress (TRB3) ↑ Insulin signalling (AKT, Foxo1)	↓ Food intake
Ropelle et al. [55]	HFD rodents	Swimming; four weeks, 60 min/day, five times/week, overload of 2.0% of the body weight Running; one session; 60 min, 10–15 m/min at a 5% inclination	↓ IKKβ, IKβα, TLR-4 ↑ Leptin signalling (JAK2/STA3) ↑ Insulin signalling (IRS1) ↑ Insulin signalling (IRS1)	↓ Food intake
Marinho et al. [59]	HFD mice	Runnning; eight weeks, 60 min, five times/week, at 60% max workload	↓ TRL-4, IKBα, TNFα, SOCS3 ↓ ER stress (PERK) ↑ Leptin signalling (JAK2/STAT3) ↑ Apoptosis (PARP1) ↑ IL-10 and Bcl2 ↑ POMC mRNA	N/A
Yi et al. [54]	LDL-receptor^−/−^ HFD mice	Runnning; six months; 30 min, 5 m/min, five times/week, 10% inclination	↓ Microgliosis	↑ Glucose tolerance ↓ Insulinemia
Laing et al. [50]	HFD mice	Voluntary running; three months;	↓ POMC apoptosis ↑ POMC proliferation ↑ Leptin sensitivity	↓ Weight gain ↑ Glucose tolerance ↑ Insulin sensitivity
Krawczewski Carhuatanta et al. [58]	HFD mice	Voluntary runnning; six weeks;	↑ Fos B immunoreactivity ↑ Leptin sensitivity (upon leptin infusion)	↓ Food intake
Silva et al. [57]	HFD mice	Swimming; 20 days, five days/week	↓ IKβα, TGF1	↓ Food intake
Wang et al. [56]	HFD Apo E^−/−^ mice	Swimming; 8 weeks, five times/day	↓ Microgliosis ↓ IL-1β, TNFα, NF-kβ	↓ Dyslipidemia

↑ Increase. ↓ Reduction. AKT, protein kinase B; ER, endoplasmic reticulum; FFAs, free fatty acids; Foxo-1, forkhead box protein O1; IκBα, nuclear factor of kappa light polypeptide gene enhancer in B-cells inhibitor; IKKβ, inhibitor of nuclear factor kappa-B kinase; IL, interleukin; IRS1, insulin receptor substrate 1; JAK2, Janus kinase 2; NF-κB nuclear factor kappa-light-chain-enhancer of activated B cells; PARP1, Poly [ADP-ribose] polymerase 1 (PARP-1); SOCS3, suppressor of cytokine signaling 3; STAT3, signal transducer and activator of transcription 3; TGF1, transforming growth factor-1; TLR-4; TLR-4, toll-like receptor-4; TNFα, tumor necrosis factor-α; TRB3, Tribbles (TRB) 3.

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
