# Peer review of "Exercise Restores Hypothalamic Health in Obesity by Reshaping the Inflammatory Network"

_antioxidants, 2023, doi:10.3390/antiox12020297_

Round 1

Reviewer 1 Report

Della Guardia and Codella evaluated animal and human evidence on the positive effect of regular exercise on obesity-induced hypothalamic inflammation/dysfunction. 

The manuscript is well written and covers a topic with interesting future perspectives. I only have a few minor comments to suggest to the authors to refine their work.

The current division into paragraphs is somewhat confusing. Paragraph headings should be more concise. Reshaping them could make reading more fluid and coherent. I suggest the following re-modulation:

1) introduction

2) Inflammatory and metabolic consequences of obesity and physical inactivity

3) Obesity-induced hypothalamic inflammation and its metabolic consequences

3.1) animal studies

3.2) human studies (if any)

4) exercise and obesity-induced hypothalamic inflammation

4.1) animal studies

4.2) human studies

5) exercise and obesity-induced hypothalamic inflammation: pathophysiological mechanisms

5.1) Systemic inflammation

5.1.1) animal studies

5.1.2) human studies

5.2) gut microbiota

5.2.1) animal studies

5.2.2) human studies

6) conclusions (with limitations of current scientific evidence and future perspectives)

Line 92: brown adipose tissue (BAT) has never been mentioned before. For the sake of completeness, I suggest adding a few brief mentions of adipose tissue types and their functional differences (maybe in line 63 where the topic was first mentioned). In addition, ensure consistency in the use of abbreviations throughout the text.

Author Response

We deeply thank the reviewer for the thorough comments provided. 

  1. The sections have been rearranged according to reviewer's suggestion. The titles of headings and subheadings were rephrased to be adherent as much as possible to reviewer's indications. In the last section, considering the paucity of evidence on humans, we decided to not discuss separately humans and animals findings.
  2. In section 2, a coincise description of brown adipose tissue physiology has been added, for completeness. 

Reviewer 2 Report

Figure 1 reports abbreviation for LP, CCl2 and other signals, which actually are not present in the Figure. Please correct.

The authors should prepare a figure illustrating the signalling pathway, the molecular effect on the metabolism and its final effect on neuronal circuit.

The abstract should be modified introducing in brief objectives and summary of the review. 

I would suggest to reconsider the title, which is maybe too specific when referring to the hypothalamic functionality.

Author Response

  1. Thanks for noticing the inaccuracies. The figure has been revised according to the comment.
  2. Thanks for the comment. In this revised version we have provided a new figure (figure 2) illustrating the effects of exercise and exercise-induced mediators on the hypothalamus, evidencing the pathways discussed in the text. The paucity of mechanistic evidence accumulated so far does not always allow to clearly define the pathways implicated. We would be willing to providing further detail or rearrange the figure under more precise indications by the reviewer. 
  3. Thanks for this comment. Abstract and title have been rearranged to meet this reviewer's thoughtful comment. 

Reviewer 3 Report

This review article covers and comprehensively present the outcome of the current literature stemming from both animal and human studies in relation to the effect of exercise intervention and prevention of obesity related inflammation and hypothalamic functionality. The review is structured nicely with information drawn from the literature put together under headings and subheadings. The writing flows well.

Although the mechanistic side of the studies has been well discussed. It would be nice to draw some attention to the exercise intensity, duration of training and the outcome- hypothalamic/inflammatory benefit. What is the current status of the literature in terms of the effect of exercise duration and intensity on these benefits?

There are number of typographical mistakes and sentence structure at few places needs modification. Please read through and remove them.

Author Response

We thank the reviewer for the favorable tones in appreciating our work. 

  1. Thanks for this helpful comment. We have added in the text all the information about frequency, intensity and duration, wherever appropriate, to complement the sections punctually describing the exercise interventions.
  2. The manuscript has been revised thoroughly, in each section, for grammar/typo mistakes. 

Round 2

Reviewer 1 Report

The authors have adequately adhered to my comments.

Author Response

We thank the reviewer for helping. 

Reviewer 2 Report

The manuscript has been substantially improved as required by the Reviewers

Author Response

we thank the reviewer for helping with the revision

Reviewer 3 Report

Authors have significantly improved the MS based on the sggestion.

Please modify as mentioned below-

Table 1-Laing et al. [50], in last column, the insulin sensitivity arrow should be upward.

Please change the figure number of last figure as 3.

Page three- line 107- aMSH – please write full form.

Author Response

Thanks for noticing the imprecision. All the inaccuracies have been corrected. Please see the highlighted sections in the text.